# Farmers' Distress Index: An Approach for an Action Plan to Reduce Vulnerability in the Drylands of India

A. Amarender Reddy [1,*], Anindita Bhattacharya [2], S. Venku Reddy [3] and Sandra Ricart [4]

1    ICAR-Central Research Institute for Dryland Agriculture (ICAR-CRIDA), Hyderabad 500059, India
2    National Institute of Rural Development and Panchayati Raj (NIRD & PR), Hyderabad 500030, India; aninditabhattacharya.87@gmail.com
3    Participatory Rural Development Society (PRDIS), Hyderabad 500048, India; sarvareddy@yahoo.com
4    Water and Territory Research Group, Interuniversity Institute of Geography, University of Alicante, 03690 San Vicente del Raspeig, Spain; sandra.ricart@ua.es
*    Correspondence: amarender.reddy@icar.gov.in; Tel.: +91-7042361439

**Abstract:** Farmer distress is a widely recognized problem in India induced by multiple causes ranging from climate variability to price volatility and the low risk-bearing ability of farmers. Tracking farmers' distress in a localized context is a prerequisite for timely action to provide sustainable livelihood options. Therefore, a field survey was conducted with 640 dryland farmers of 10 sub-district units from two states in India with the aim to identify the major indicators based on seven dimensions of distress and to construct a multidimensional Farmers' Distress Index (FDI) at the farmer and sub-district levels. The FDI was built with seven dimensions of distress: exposure to risk, adaptive capacity, sensitivity, mitigation and adaptation strategies, triggers, psychological factors, and impacts. The study developed a broad-based FDI which can be used as a planning tool that can address the causes of farmers' distress and also evolve measures to tackle those causes. Based on the result, the study recommends a location-specific distress management package based on various dimensions of the FDI. The paper also suggests an upscaling strategy to identify and prioritize the highly distressed farmers as well as sub-district geographical units by tracking a few sets of variables.

**Keywords:** distress indicators; agrarian risks; resilience measure; distress management; India; intervention

## 1. Introduction

India has the highest share of total cropped area (66%) [1] among the rainfed agricultural countries in the world and contributes to half of the country's total food production [2]. The government of India identified 177 districts out of 718 districts as predominantly rainfed based on the extent of the rainfed area [3,4]. Drylands are referred to as low rainfall (less than 75 cm annually) areas where mixed farming is practiced with a mix of drought-tolerant crops, with the rearing of small ruminants representing the primary income source [5]. The rainfed agricultural system is hazardous as it heavily depends on erratic rainfall and human interentions such as investment in natural resource management and land use management [6,7]. Since the 1990s, after the country's liberalization, government regulations have been relaxed, and greater participation by private entities is also restricted. Thus, the rainfed farmers are becoming further exposed to price risks as they heavily depend on markets for purchasing inputs and selling outputs, unlike subsistence farming [8]. Together, these stressors have the immense potential to threaten the sustainability of the livelihoods of a large number of agricultural households [9]. In addition to the impending climate change, drylands are also affected by various other inherent biotic and abiotic limitations such as water scarcity, prolonged droughts, the late onset of monsoons, wind and soil erosion, and pests and disease infestations [10].

The challenges faced by rainfed agriculture are interrelated with the various Sustainable Development Goals (SDGs) established by the United Nations in 2015. This includes

SDG 1 (no poverty), SDG 2 (zero hunger), SDG 13 (climate action), and SDG 15 (life on land) [11]. Alleviating farmers' distress is a prerequisite for the sustainable development of rainfed areas, as distressed farmers overexploit and underinvest in natural resources such as water and land. As the land challenge underlying the nation's agricultural crisis, land resources play a crucial role in farmers' distress. Thus, once farmers' distress is alleviated, farmers tend to enhance their soil and water conservation investments, leading to sustainable livelihoods [1,3,4]. Climate change projections remark that frequent droughts, prolonged dry spells, and high temperatures coupled with fluctuations in prices expose farmers to high risks, in severe cases resulting in the suicides of farmers [12]. Earlier studies by Krishnamurthy [13] or from European Cooperation in Science and Technology (COST) [14] observed that the severe crisis in dryland agriculture in the past century has resulted in increased levels of poverty and low investment in both human and physical capital, which reinforce low agricultural productivity and low incomes.

Over several decades, the neglect of rainfed areas by public and private sectors and farmers themselves has resulted in meager accumulated capital to conserve natural resources [15]. The high exposure to natural hazards and low accumulated capital coupled with the small land-holding size of the majority of the rainfed farmers have led to low and fluctuating farm incomes in most developing countries of Asia and Africa [16]. These farmers in tropical countries, including India, have limited resources and capacity to cope with these shocks [17] and are experiencing severe hardship due to their low adaptive capacity [18]. Likewise, they cannot make appropriate decisions as they face multiple constraints while adjusting to these shocks [19]. Furthermore, agricultural insurance that depends on government subsidies eases the burden on farmers who experience crop failure [20].

Besides these factors, farmers also face social and psychological challenges such as inequality, marginalization and socioeconomic deprivation, isolation, and depression [21]. Thus, any exposure to shocks severely impacts the sustainability of their livelihoods [22]. According to the literature, distress or vulnerability is the degree of susceptibility of a natural ecosystem or socioeconomic system [23], and agricultural vulnerability is generally defined as the probability of loss and damage of an agricultural system [24]. The vulnerability of the agricultural households in dryland regions depends on external (intensity of disaster and harm inflicted) as well as internal factors (differential capacity of households) (Figure 1) [25]. Therefore, this is referred to as the "dualistic structure of vulnerability" [26]. Exposure to hazards and natural calamities such as droughts has different impacts on vulnerability depending on the severity of the hazard and the households' adaptive and coping capacity and livelihood options [27,28]. Households with a lower capacity are comparatively at more risk, and vice versa.

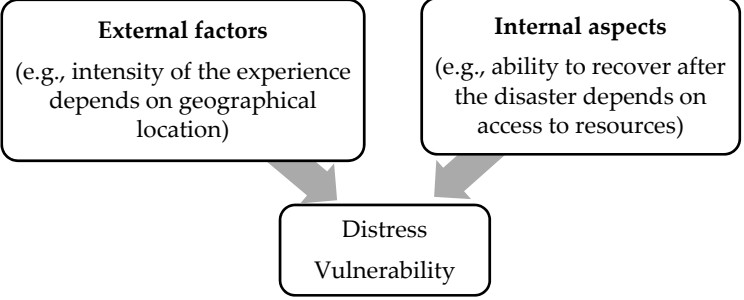

**Figure 1.** Internal and external aspects of livelihood distress [29].

The model illustrated in Figure 1 indicates that rainfed farmers are increasingly exposed to risks, which may lead to farmers' distress depending on the causes, capabilities, coping, and adaptation strategies. Thus, increasing resilience (the capacity to recover quickly from difficulties) to exposure to risk is of primary importance. Building resilience represents the quest to improve the quality of life and well-being of people, especially when

exposed to adverse conditions [30]. In recent years, antifragility, which means the capacity to gain from disorders that are gaining momentum, has been introduced. However, to become antifragile, one needs to have some other capabilities, such as high investments in human and physical capital, that are absent in most households living in the backward, rainfed areas in India [31]. If antifragility is not possible, resilience is practically achievable for the vulnerable, resource-poor, women, and, most importantly, small landholding farmers [32]. Resilience also depends on government policy, which evolves continuously depending on the local socio-economic and ecological context [33]. A sound resilience policy has to be implemented to address various dimensions of vulnerability [34].

In this paper, resilience or antifragility and vulnerability or distress are used as synonyms as both of the former terms have a similar meaning of "overcoming difficulties" from farmers' perspectives. Similarly, vulnerability and distress have the same meaning of "being open to harm or damage" [31].

## 1.1. Objectives of the Study

The various indicator approaches developed worldwide do not indicate any fixed method to measure agriculture's vulnerability to climate variability [20]. To target distressed farmers through policy intervention, the government should quantify the farmers' distress and identify geographies and farmers with severe distress. However, there is no standard measure of farmers' distress apart from the number of suicides, which is a post-mortem indicator and cannot be used as a policy tool [35]. Therefore, there is a need to develop an index including the complexity and multidimensionality of distress [36,37]. Although some district-level indices quantify sustainability, risk, and vulnerability [38,39], there is no standard index available for measuring farmers' distress by taking the multidimensionality of farmers and sub-district levels into account. A few studies are available for Mozambique [40], Bangladesh [41], and Trinidad and Tobago [42], while a study was produced for the Himachal region of India [43], but they quantified a general vulnerability index for a particular context and lack scalability. As an improvement over the past studies, this paper focuses on identifying and screening indicators to construct a multidimensional Farmers' Distress Index (FDI) at the farmer and sub-district levels [44]. This index covers seven dimensions of distress, integrating exposure to risk, sensitivity, adaptation and coping strategies, adaptive capacity, triggers, socio-psychological factors, and impacts to help in the development of intervention points based on various dimensions of the index [35]. The FDI developed in this paper will help in identifying severely distressed farmers and sub-districts and prioritize actions [27]. Thus, the FDI is a tool for planning, monitoring, and also executing policy interventions to reduce the misery of distressed farmers.

Composite indexes are increasingly relevant as measurement and monitoring tools in public policy rather than single variable indicators such as indebtedness (a symptom of the farmer's distress rather than the cause). The complexity of farmers' distress raises the need to synthesize complex phenomena by considering a multidimensional index by integrating causes, adaptive capacities, and impacts [45,46]. Furthermore, one of the principal advantages of using a composite index is its simplicity and capacity to be easily understandable to end-users and non-experts; furthermore, it is unbiased, meaning that the wider public can be convinced easily of the utility of these indices in acting [47,48].

## 1.2. Research Questions

In the process of developing the index, the research questions raised in this study were the following: (i) What dimensions need to be included in the construction of a multidimensional FDI? (ii) Is a district homogenous enough to rely on district-level indices, or is a sub-district level index required for better targeting? (iii) How can we scale up the action based on the FDI to cover all sub-districts?

The following section deals with the sampling framework, study area, and methodology, followed by selection and screening of the indicators and the construction of the FDI. Section 3 presents the results and discusses the different dimensions of the FDI and the

use of the index for the prioritization of geographies and policy interventions. Section 4 concludes the work with policy implications.

## 2. Materials and Methods

### 2.1. Study Area

A pilot study was conducted in the Andhra Pradesh and Telangana states of India (Figure 2)—two adjacent states which are drought-prone and reported the highest number of farmer suicides in recent years. These two states are among the five states with the most significant number of farmer suicides in India, constituting 25% (628 cases) and 12% (491 cases) of farmer suicides, respectively [49]. The states cover 162,975 and 112,077 km$^2$ and have populations of 52.5 million and 39.8 million, respectively, according to the 2021 census. Their economy is driven by small landholder agriculture, with about 86% of the farmers having less than 2.5 acres (=1 ha) of land.

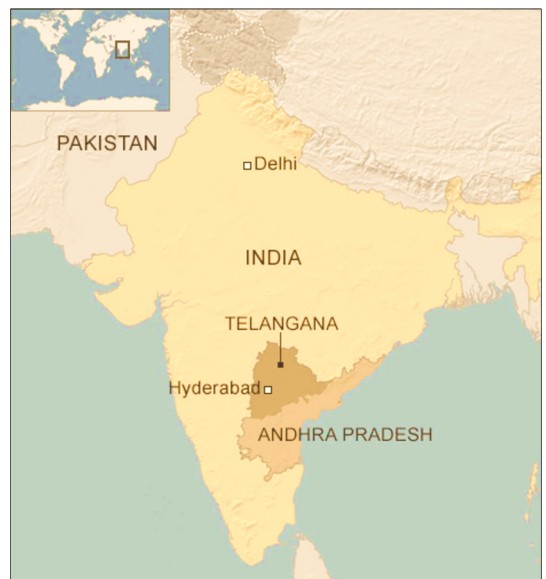

**Figure 2.** Map showing the location of study area (Telangana and Andhra Pradesh).

### 2.2. Sampling Framework

The selection of the districts was based on the high number of suicides, as it is the most commonly cited measure for farmers' distress [49], and they are being the most prominent areas under dryland agriculture (with equal weighting). Andhra Pradesh is subdivided into 13 districts, and Telangana is sub-divided into 31 districts. From each state, the two districts with the highest number of farmers' suicides during the last decade and the maximum area under dryland agriculture with equal weighting were selected. Then, two of the highest drought-prone *mandal*s ('*mandal*' means the administrative unit below the district level; administratively, each district is divided into 40–50 *mandal*s, and in some states, these sub-district administrative units are called blocks) from each district in Telangana and three *mandal*s from each district of Andhra Pradesh were selected for an intensive field survey based on the extent of area under dryland agriculture. From each state, 16 villages from the selected *mandal*s were selected, again based on the highest drought-prone area (each *mandal* has approximately 30–40 villages). From each village, 20 farmers were selected randomly for an intensive field survey. The sample comprised 4 districts, 10 *mandal*s, 32 villages, and 640 farmers. The data were collected from December 2020 to February 2021. The detailed sampling framework is shown in Figure 3.

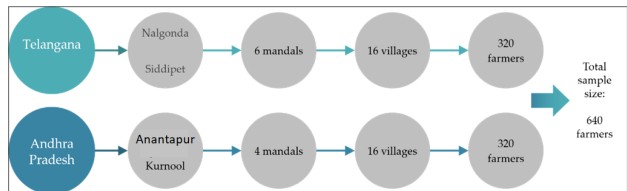

**Figure 3.** Sampling framework.

### 2.3. Identification of Indicators

The farmers' distress indicators were collated and screened through an extensive review of the published literature in peer-reviewed journals and based on focus group discussions with key informants. The final questionnaire, which includes both open- and close-ended questions, was developed after discussion in focus group interactions regarding probable distress indicators and was pre-tested, refined based on feedback, and ultimately included only 123 indicators. The identification of the indicators was conducted through the particular process shown in Figure 4. The indicators used in this study were classified based on seven dimensions of vulnerability, which is an improvement over an earlier study [17,50]. The seven dimensions were exposure to hazard, sensitivity, adaptive capacity, mitigation and adaptation strategies, triggers, socio-psychological aspects, and impacts.

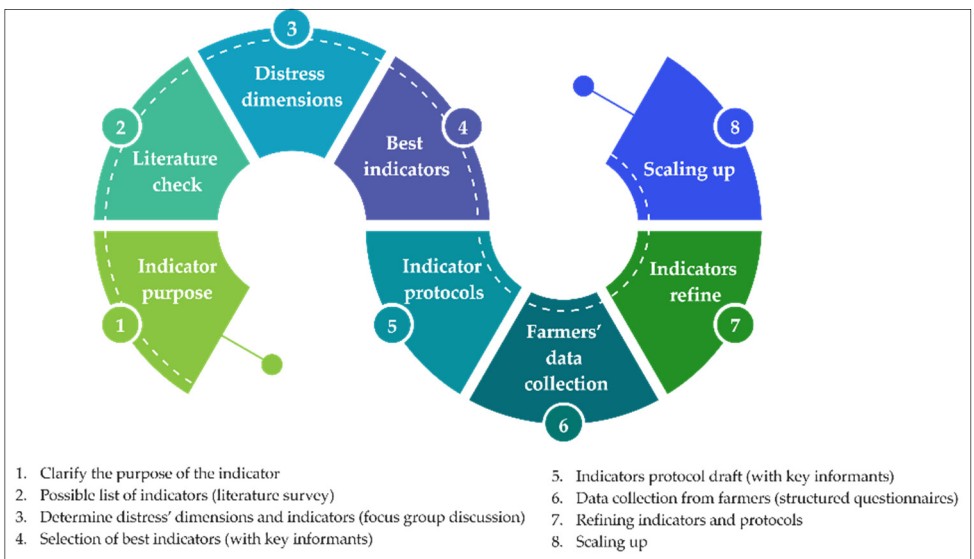

**Figure 4.** Indicator selection process.

In the literature regarding agrarian distress, there are many references to vulnerability [51–58]. Although there are differences in definitions, all approaches to agricultural vulnerability broadly include exposure, sensitivity, and adaptive capacity [17]. Here, exposure is defined as the nature and degree of a system's exposure to climatic variations, and sensitivity is the degree to which a system is affected, either adversely or beneficially, by exposure to drought/floods, for example. Adaptive capacity is crucial to modify exposure to risks, absorption, and recovering capacity from the losses stemming from exposure. Otherwise, adaptive capacity is defined as the propensity or predisposition to be adversely affected. Thus, to reduce vulnerability stress, it is essential to decrease sensitivity and strengthen the adaptive capacity of local communities. The adaptive capacity varies between different contexts and systems and is closely linked with infrastructural, institutional, community, social, political, demographic, economic, educational, health, technological, and cognitive factors.

Farmers follow both mitigation and adaptation strategies against exposure to hazards. Recent research highlights synergies between the mitigation and adaptation strategies followed by farmers [59]. Mitigation comprises all human activities aimed at reducing adverse events such as droughts and floods through the construction of check dams, percolation tanks, etc. Adaptation strategies refer to any adjustment performed by the farmers or farming community in response to exposure to hazards, such as droughts, to moderate harm or exploit beneficial opportunities [60]. This paper considers both mitigation and adaptation strategies together as one dimension of farmers' distress (Figure 5). A trigger event is an occurrence that causes severe distress, such as the failure of a borewell after the investment of vast amounts of money in digging the borewell [61]. Social and psychological factors are significant provocations for extreme events such as farmer suicides. The impacts are the ultimate result of all the above indicators regarding farmers' incomes, indebtedness, etc. The central focus of the FDI [62] is to look into all dimensions of farmers' distress and quantify each dimension [63].

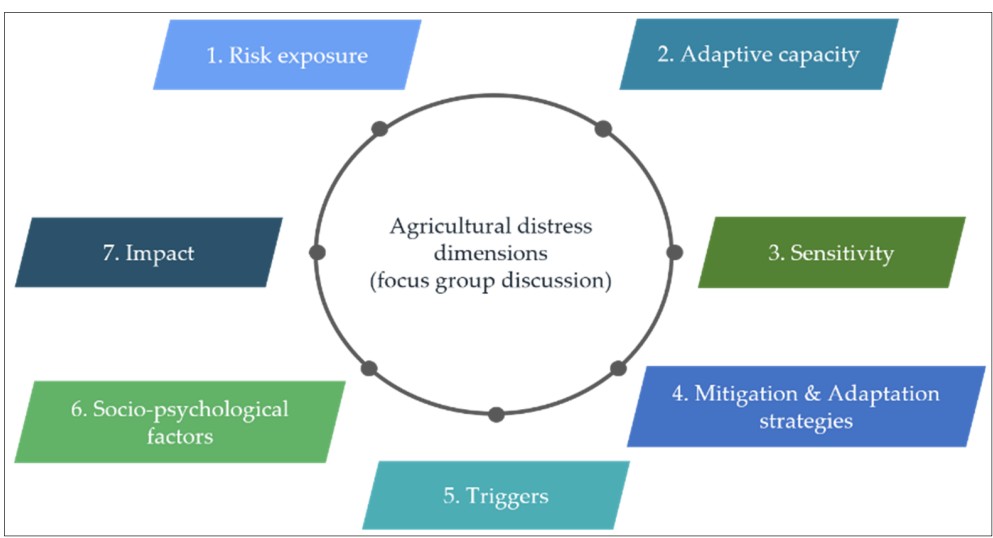

**Figure 5.** Major dimensions of agricultural distress.

### 2.4. Screening of Variables for Final FDI

In this study, we aimed to identify and screen indicators of farmers' distress to develop a composite FDI able to identify the sources, forms, and depth of vulnerability specific to the context to design resilience measures (Table 1). This method of parameters is a new approach to assess farmers' vulnerability. As the literature on farmers' vulnerability is very limited in India, this index can help evaluate this issue at the sub-regional level in the country [17].

**Table 1.** Systematic design of the research FDI (Farmers' Distress Index).

| Research Purposes | Analysis Tools | Data | Results |
|---|---|---|---|
| Identification of major indicators of farmers distress | Descriptive statistics using CREAM criteria [64] | Primary data | FDI |
| Identification of geographies (sub-district level) with severe farmers' distress | FDI scaling | | Farmers' vulnerability mapping |

The study not only develops an FDI but also decomposes the index into seven dimensions. As a result of this attempt, the study recommends prioritizing the interventions to alleviate farm distress. Furthermore, by aggregating the FDI at the sub-district level, the paper presents a methodology to develop the FDI at the sub-district level to categorize

high, medium, and low distress areas for prioritizing fund allocation, with more funds given to areas with a high FDI.

### 2.5. Collection of Primary Data

Primary data were collected from sample households using a structured questionnaire with open- and closed-ended questions in the crop year 2020. The study was undertaken at the farmer level to collect data on the seven dimensions of farmers' distress. In each village, census data were collected from government departments; then, twenty households were randomly selected in each village for an intensive survey.

The data from all sample farmers were collected for 123 variables. However, while screening indicators, a modified version of the CREAM criteria [64] was used: the performance indicators should be clear, relevant, economic, adequate, and monitorable (CREAM). The rating was conducted on a scale of 0 to 2, where "0" indicated a low score and "2" indicated a high score in the relevant performance criteria. The simple total score of all six criteria was used to select indicators (Table 2). The variables used for index development were tested for correlations with other variables at 95% confidence, and more highly correlated variables were removed while calculating the FDI.

**Table 2.** Rating of variables of indicators.

| Variables | Scores | | | | | | Total Score | Selected Variables of Each Indicator with the Highest Score |
|---|---|---|---|---|---|---|---|---|
| | **A** | **B** | **C** | **D** | **E** | **F** | | |
| Indicator | | | | | | | | |

Note: A—Clear meaning; B—Data are easily available; C—Less effort in data collection, and the data do not require expert analysis; D—Sufficiently representative for the total of the intended results; E—Tangible and observable; F—Difficult to quantify but very important (proxy indicator).

After screening all the indicators in all seven dimensions, out of 123 indicators, only 50 were included in calculating the composite FDI (details shown in Supplementary Table S1). Of the total of six indicators for risk exposure, three were selected; out of 31 indicators for adaptive capacity, 14 were selected; out of 10 indicators of sensitivity, 6 were selected; out of 39 indicators of mitigation and adaptation strategies, 13 were selected; out of 14 indicators of triggers, 4 were selected; out of 14 indicators of social and psychological factors, 6 were selected; and out of nine impact indicators, four were selected.

### 2.6. Tools and Techniques

The indicators were measured on different scales—e.g., some were numbers or percentages, and others were indices. Therefore, they were normalized to a range between 0 and 1 (Table 3). For indicators that decreased distress, the values were transformed so that the derived indicator had a positive association with FDI (e.g., 100 minus the indicator value in the case of percentage units).

**Table 3.** Calculation of indices.

| Standardization of indicator formula | $Index_{sw} = (S_w - S_{min})/(S_{max} - S_{min})$ | Index Scale |
|---|---|---|
| Major dimensions formula (7 dimensions) | $M_w = (\sum_{i=1}^{n} Index_{swi})/n$ | 0 = least vulnerable to |
| Overall index formula (comprising 50 variables) | $FDI = (\sum_{i=1}^{7} W_{mi}M_{wi})/(\sum_{i=1}^{7} W_{mi})$ | 1 = most vulnerable |

$S_w$ is the original indicator value for the household. $S_{min}$ and $S_{max}$ are the minimum and maximum values, respectively. $M_w$ is one of the major dimensions of the seven dimensions for measuring distress. $Index_{swi}$ is the indexed indicator for households. n is the number of indicators for each major component. $W_{mi}$ is the weight of each major dimension. $M_{wi}$ is the average value of each major dimension.

The distress level is scaled from least vulnerable, with a low index value, to most vulnerable, with the highest index value [65]. The indicators and their weights were assessed using multiple techniques such as expert opinion and literature review, regressing each variable with farmers' distress indicators, such as farm debt from informal sources (such as

money lenders) with high interest rates. However, in this paper, we assigned equal weight to all the variables while constructing all seven dimensions of distress as this removes subjectivity and makes the index easy to upscale. The equal weighting was preferred because it makes it easy to calculate the index for the administrators/implementers and to avoid pressures from political leaders to engage in manipulation by changing weights arbitrarily for the inclusion of their political constituencies in high-priority lists to obtain more funding, for example, while scaling up the index across India.

*2.7. Robustness Check (Out-of-Sample Validation)*

Testing the robustness of multidimensional composite indicators such as the FDI is critical for the developed indicator to be scalable across a wide variety of geographies and socioeconomic settings. Undeniably, 'traditional' or otherwise, robustness analysis in any form may act as a quality assurance tool [66]. However, one of the first points stressed in the OECD's Handbook is that one cannot interpret an assessment of robustness to validate a 'sensible' index [67]. Instead, it creates a sound theoretical framework that determines whether the index is sensible.

In this paper, the FDI is mainly developed as an administrative tool to identify highly distressed farmers and sub-districts by using equal weightings for all seven dimensions. The robustness was checked for scalability across geographies and farmer groups. For this purpose, we constructed FDIs for the twenty households who reported farmers' suicide and twenty well-off households in non-study areas (out-of-sample validation). The FDI was very high (>0.95) for the former group, while it was significantly lower (<0.50) for the latter group at a 95% confidence interval, indicating the robustness of the FDI index.

## 3. Results and Discussions

### 3.1. General Profile of the Households

Several studies elucidated that the preponderance of marginal holdings is increasing due to great demographic pressure and land segregation [68–70]. In the present study, the average operational land holding of the sample households was 3.8 acres (1.52 ha), among which 90% of the area was under rainfed farming. Rainfed farming exposes farmers to various types of risk and uncertainty such as droughts, dry spells, biotic and abiotic stresses, and a shortage of water for rearing livestock, among others. The significant sources of household income were cultivation, agricultural labor, casual labor in non-agricultural sectors, and salaried employment in other non-farm sectors. The average household income was around INR 97,000 (USD 1300) per annum. The lack of primary education among the farmers was visible. The household members' age distribution indicated that there is a higher proportion in the productive age group, which indicated the higher potential of economic activities for the region. However, the severe crisis in agriculture is ruining this potentiality and making the lives of farmers extremely vulnerable.

### 3.2. Measuring Multiple Dimensions of Distress

This paper attempts to develop an FDI that can be used across temporal and spatial dimensions without losing the local context. It is recognized that there are multiple drivers of vulnerability at the local level that can be used to assess the extent and depth of farmers' distress (Table 4). The major dimensions of the FDI are:

- Exposure (natural disaster), which is the magnitude and duration of the population's exposure to distress.
- Adaptive capacity (socio-demographic profile, livelihood strategies, social networks), which denotes the household's or farmer's ability to resist and adapt to distress.
- Sensitivity (health, food, water), which is the degree to which a household is affected by distress.
- Mitigation and adaptation strategies, which are aimed at tackling the causes and minimizing the possible negative impacts of exposure to risk.

- Triggers, which are events or situations that provoke farmers to take extreme steps, such as suicide [61].
- Psychological factors, which are essential drivers of severe depression, isolation from society, etc.
- Impacts, which are the results of agrarian distress.

**Table 4.** Final variables selected after screening of the multiple dimensions of farmers' distress.

| Dimension | Components | Variables |
|---|---|---|
| Exposure | | Flood/cyclone, drought, low output price |
| Adaptive capacity | Indices of socio-demographic profile | Sex, caste, religion, family size, elderly population, disability, sex ratio, educational status of the head, illiteracy, dependency ratio |
| | Socioeconomic assets | House value, gold value, total land |
| | Livelihood strategies | Savings, income, Simpson Diversity Index (SDI), household expenditure |
| | Social networks | Membership in SHG, cooperative, agricultural cooperative, local cooperative |
| | Agricultural activities | Agricultural input, income from agriculture, profit, production cost, total owned land, total leased in the land, profit/acre, rainfed operational |
| Sensitivity | Water | Provision of rainwater harvesting, failure of borewells |
| | Health | Health expenditure, likely health expenses, chronic illness |
| | Food | Food expenses |
| | Infrastructure | Road |
| | Children | Children enrolled in private schools, likelihood of withdrawal of children from schools |
| | Finance | Indebtedness through informal source |
| Mitigation and adaptation strategies | Farmer's initiatives | Reduced cropped area, land kept fallow, low input use, reduced household expenses, borrowing from relatives and friends, borrowing from money lenders, migrated out as a casual laborer, participation in MNREGA, postponed health treatment, postponed marriages, sold livestock animals, engaged in animal husbandry, sending women for domestic work |
| | Benefits from government schemes | From any formal institution, *Rythu Bandhu*, SHC, KCC, old-age pension, health scheme, child education, insurance scheme |
| | Adaptation strategies | Use own savings, migrate to other places, change cropping pattern, sale of assets, reduce expenditure on food, take children out from school, borrowing, depending more on non-farm employment, bonded labor, take support from local government |
| | Constraints in adaptation measures | Low education level, lack of access to information, inadequate capital, lack of extension services, land not suitable, lack of irrigation, non-availability of labor, lack of quality inputs |
| Triggers | | Health expenses, marital disputes, chronic illness, children's marriage dowry, educational expense, other marriage expense, unemployment, lack of alternative income source, frequent pest and disease attack, outbreak of livestock disease, lower price of output, high farm expenses, crop failure, debt from informal sources |
| Psychological factors | Social issues | Objection to the participation of women, catastrophe incidence in last five years, negative comment from society |
| | Change in social position | Feeling isolated |
| | Mental harassment | Serious issue with society |
| | Family burden | Unable to fulfill familial responsibility, lack of moral support, major family issue |
| | Deterioration of economic status | Worried about financial distress, family issues regarding the deterioration in economic status |
| | Behavioral change | Addiction to smoking, alcohol, or drugs |
| | Loss of self-confidence | Chronic stress, loss of pleasure in economic activities, thought of ending life |
| Impact | | Reduced income, increased indebtedness, shortage in food consumption, purchased food from outside, increased poverty, deteriorating health, social stigma, sale of livestock/poultry, facing distress in last five years |

Source: Farmer's survey.

### 3.2.1. Exposure to Risk

Risk exposure is the level or the magnitude and duration of exposure to disaster/hazard/risk for farmers' households [61]. The components of exposure comprise variables such as floods or cyclones, drought, and low output prices. According to the focus group interactions and data analysis from 640 farmers in the study areas, these disasters occurred much more frequently during the past five years and were the reason behind the losses of income. Exposure involves climate variables as well as price factors. The indicators of the three selected variables in the four study districts are given in Figure 6.

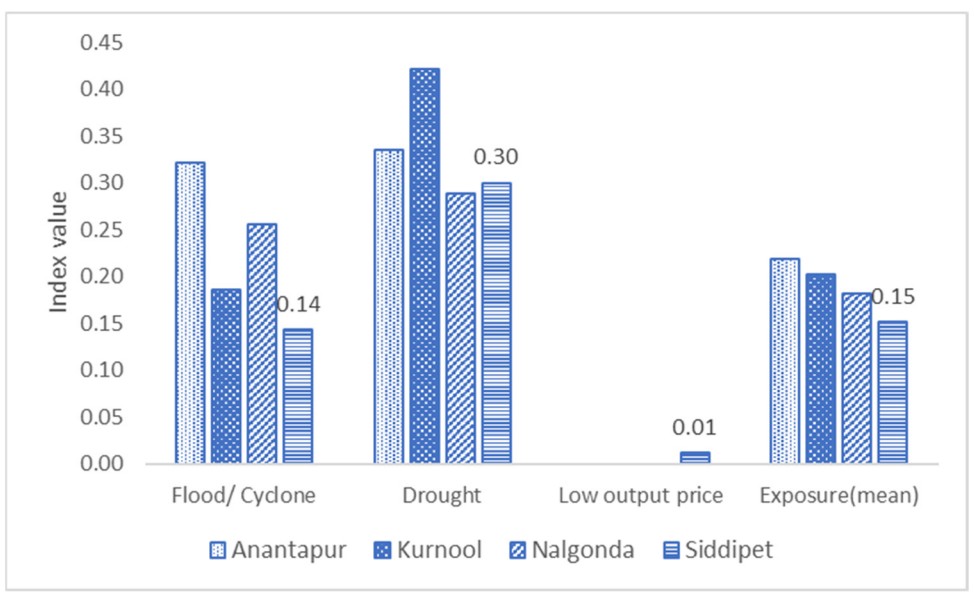

**Figure 6.** Indexed values for the major indicators of exposure.

Here, *Anantapur* district depicts the highest value for disasters as this district has been highly affected by weather calamities and other disasters. This area has received low rainfall year after year as it lies in a rain shadow region and is thus more prone to experiencing severe droughts.

### 3.2.2. Adaptive Capacity

This indicator represents households' capacity to cope with distress conditions. Sociodemographic profile (which includes variables such as caste, sex ratio of family, educational status of the head, and dependency ratio), socioeconomic assets (house value, total own land), livelihood strategies (total savings, Simpson Income Index, Simpson Diversity Index—cropping pattern), social networks (membership in SHGs/cooperatives), and agricultural activities (income from agriculture, profit, and rainfed area) are components of the adaptive capacity index.

Caste (forward caste = 2; backward caste = 1; scheduled caste = 0) and the educational status of the head of the household (years of education) were assumed to be inversely associated with FDI among the socio-demographic variables. Meanwhile, among the livelihood strategies, total savings, the Simpson Income Index, and the Simpson Diversity Index (SDI) of cropping patterns were assumed to have an inverse association with FDI based on the focus group discussions. Income from agriculture and profit were also assumed to be inversely associated with FDI. The remaining variables were assumed to have a positive association with FDI (Figure 7).

Farmers with more diverse livelihood strategies have a higher adaptive capacity than less diversified households. Households with more social will have a higher adaptive capacity, and households having a share of agricultural income in their total income showed a negative association with FDI [71].

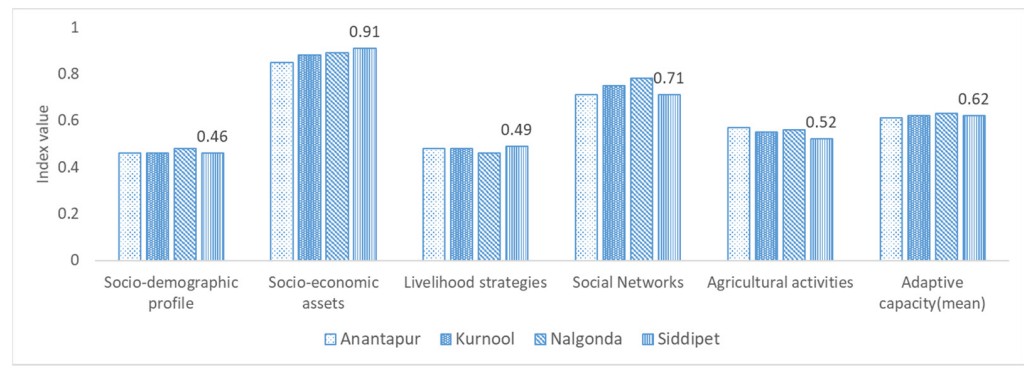

**Figure 7.** Indexed values of major indicators of adaptive capacity.

The components of adaptive capacity included socio-demographic profile, socioeconomic assets, livelihood strategies, social networks, and agricultural activities. Assets and social networks play a more significant role, while the demographic profile and livelihoods strategies contribute less to adaptive capacity. The mean adaptive capacity index values did not vary much among districts, which may be due to the districts' similar socioeconomic settings and historical backgrounds (Tables S3–S7).

3.2.3. Sensitivity

Under the sensitivity dimension, six major indicators—i.e., water (borewell failure), health (health expenditure), food (food expenses), infrastructure (roads), children (withdrawal of children from schools), and finance (indebtedness from informal source), which are very basic necessities and socially sensitive components for any community—were included. Except for infrastructure (roads), all other indicators are positively associated with FDI (Table S1). All four districts exhibited low index values for health, infrastructure, and finance, which shows their low contribution to farmers' distress (Figure 8 and Table S8). *Nalgonda* displayed the highest vulnerability in terms of water, as the district's water table is very intense and overexploited due to mismatched cropping patterns in favor of water-intensive crops such as paddy; this resulted in vast numbers of borewell failures. Government support programs such as the Public Distribution System (PDS), primary health centers, and government schools reduced the sensitivity of the households to distress.

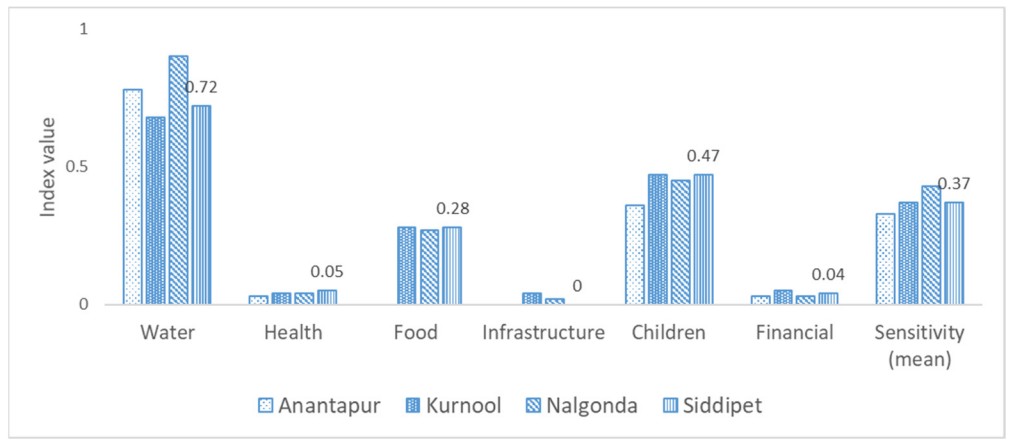

**Figure 8.** Indexed values of sensitivity indicators.

However, water stress, educational expenses for children—as more parents are sending children to private schools with exorbitant fees beyond their capacity—and food expenses remain a source of distress and sensitivity [13].

### 3.2.4. Mitigation and Adaptation Strategies

Due to the inherent nature of high-risk agriculture, farmers follow different mitigation and adaptation strategies [72]. These indicators are grouped into two categories: the first category includes strategies followed by farmers themselves, such as reduced cropped areas in the event of drought, reduced household expenses, borrowing from money lenders, and selling of livestock animals; the second category includes measures taken by the government to help farmers. Some schemes are mainly targeted to distressed situations and farmers, and others are more general safety nets or income support programs. Most farmers are beneficiaries of government schemes, and almost all landholders benefited through the *Rythu Bandhu* scheme. The state government introduced the scheme, and it is a direct benefit transfer program led by the government to transfer INR 10,000/acre/annum. Some farmers are beneficiaries of an old-age pension scheme through an insurance scheme, and some participate in employment guarantee schemes such as the Mahatma Gandhi Rural Employment Guarantee Act (MGNREGA). Sometimes, farming households migrated to other places in search of income and employment opportunities or temporarily shifted to non-farm employment such as casual labor in construction activities. Farmers faced some constraints while adopting these mitigation and adaptation strategies, the most important of which are low education level and awareness, inadequate capital, and irrigation facilities.

However, the lowest index value in the case of farmers' initiatives in Figure 9 (and Tables S9–S12) shows that farmers' initiatives to mitigate risks by following their strategies, such as reducing cropped areas, reducing household expenses, borrowing from money lenders, and selling livestock, have much greater importance than the government support in reducing farmers' distress. The contribution of constraints to adaptation to the FDI was outstanding. Hence, removing constraints by increasing awareness about mitigation and adaptation strategies, providing financial assistance, and increasing water sources for crops and livestock will reduce distress.

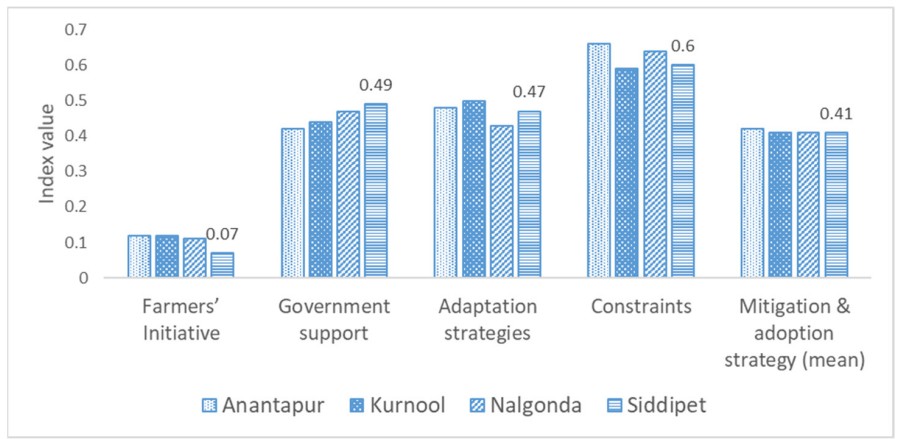

**Figure 9.** Indexed values for the indicators of mitigation and adaptation strategies.

### 3.2.5. Triggers

Farmers' exposure to hazards, sensitivity, adaptive capacity, and mitigation and adaptation strategies set the conditions for farmers' distress at the local level in favor of or against distress. However, trigger events such as ill-health, unexpected household expenditures, or unemployment forced farmers to take extreme steps such as suicide or led to psychological disorders. It is thus important to determine the triggers contributing to the aggravation of severe farmer distress. After discussing with farmers in focus group interactions, the study team presented some trigger indicators that are the starting point for severe distress. The most significant and immediate factors leading to agrarian distress are chronic illness, dowry expenses (an amount of property or money brought by a bride to her husband on their marriage) during a daughter's marriage, unemployment, and crop failure

(Figure 10 and Table S13). These triggers cause farmers to take some drastic decisions such as borrowing from money lenders at very high interest rates, clashes within families, hunger, diseases, and, in extreme cases, attempted suicide. Although chronic illness and a daughter's marriage expenses do not contribute much to distress, unemployment and crop failures are major factors.

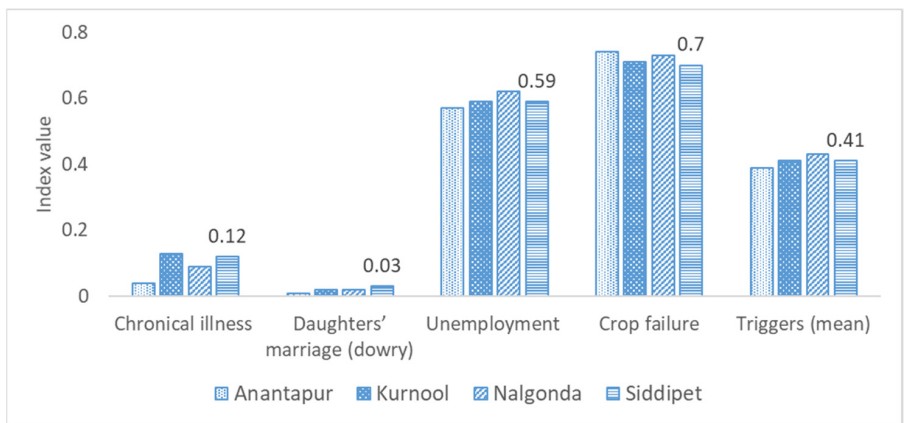

**Figure 10.** Indexed values for the indicators of triggers.

### 3.2.6. Change in Social–Psychological Aspects

Several studies have shown that agrarian distress is related to farmers' social and psychological conditions [73–76]. Thus, the present study also focused on social and psychological factors and the behavioral change of farmers concerning society and other household members. Out of 14 indicators, only five were selected for intensive tracking and included in the FDI (Figure 11 and Table S14). These five indicators were family burden (unable to fulfill family responsibilities), deterioration of economic status (worried about financial distress), behavioral change (addiction to smoking, alcohol, or drugs), and loss of self-confidence (facing chronic stress). The findings show that the most common factor that depressed the farmers psychologically was family burden. The feeling of inability to fulfill family responsibilities and the lack of moral support from other family members made the farmers psychologically upset to such an extent that they would take extreme steps such as absconding from the village. Furthermore, mental harassment of farmers due to loan repayment or some severe conflicts with society has been reported. These two factors are also responsible for farmers losing their self-confidence and some behavioral changes such as addiction to alcohol. Therefore, all these factors together make farmers more vulnerable.

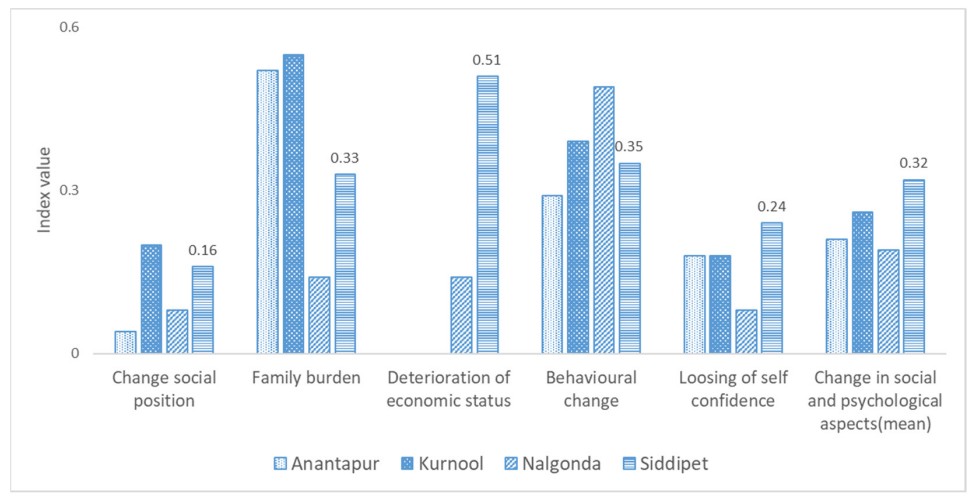

**Figure 11.** Indexed values for the indicators of changes in social and psychological factors.

### 3.2.7. Impact

The factors discussed in the previous sections have a synergistic and mutually reinforcing effect on farmers' incomes and indebtedness. The low and highly fluctuating income from agriculture has a detrimental effect on farm investments and forces the farmers to become trapped in debt, thus increasing indebtedness. This indebtedness of farmers is perhaps the leading determinant of agrarian stress as it quickly corrodes and destroys the farmers' livelihoods and resilience [77–83]. A severe crisis in agriculture begins with the failure of crops and sets off a vicious cycle of socioeconomic impacts such as the erosion of assets and income decline, indebtedness, poverty with hunger and malnutrition, and a deterioration in the standard of living, thus increasing the vulnerability of poor farmers [84]. The duration and depth of adverse impacts are reflected in four indicators: reduced income, increased indebtedness, increased poverty, and distress faced in the last five years (Figure 12 and Table S15). Here, the distinction between "reduced income" and "increased poverty" is that the former reduces income levels irrespective of poverty status while the latter is a severe form of income reduction below the poverty line. The study shows some common trends in different districts with a higher contribution of reduced income and increased indebtedness. Of all the districts, impacts were most severe in *Siddipet* district, mainly due to distress in the last five years.

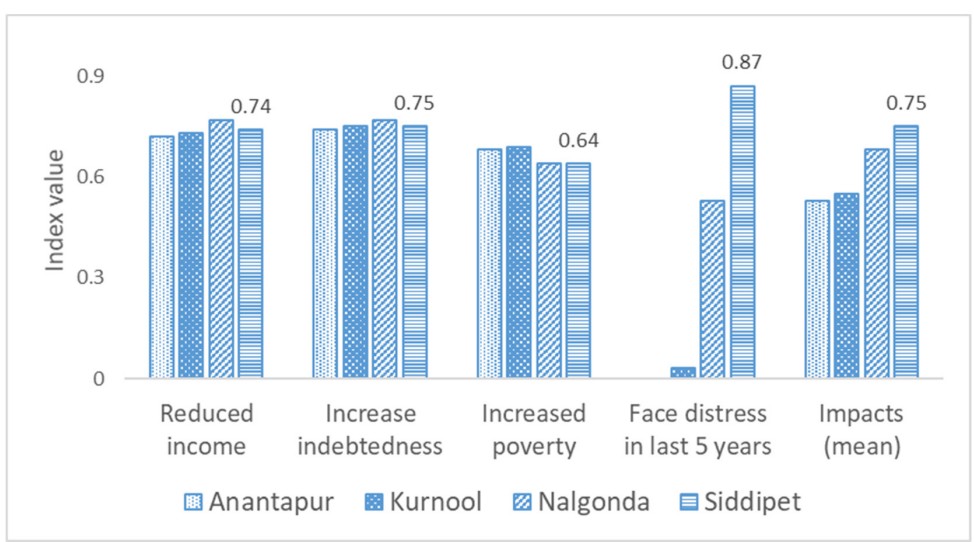

**Figure 12.** Indexed values for the indicators of various impacts.

### 3.3. Dimension of Farmers' Distress and the FDI

The empirical results after analyzing all the components of each of the seven dimensions of the FDI after normalization with values of 0 (less vulnerable) to 1 (more vulnerable) are presented in Figure 13. The results reveal that all the districts are extremely vulnerable in terms of adaptive capacity, mitigation and adaptation strategies, triggers, and impact indicators. This indicates a need to enhance the adaptive capacity of farmers, popularize mitigation and adaptation strategies, remove trigger events, and reduce adverse impacts through a multiagency approach by involving farmers, the local community, and the government in partnership mode.

The overall FDI value in the districts ranged between 0.396 and 0.432, indicating moderate agrarian distress in all four districts. The major vulnerability components for the FDI are presented in Figure 13, illustrating the features that contribute more to the vulnerability of the areas.

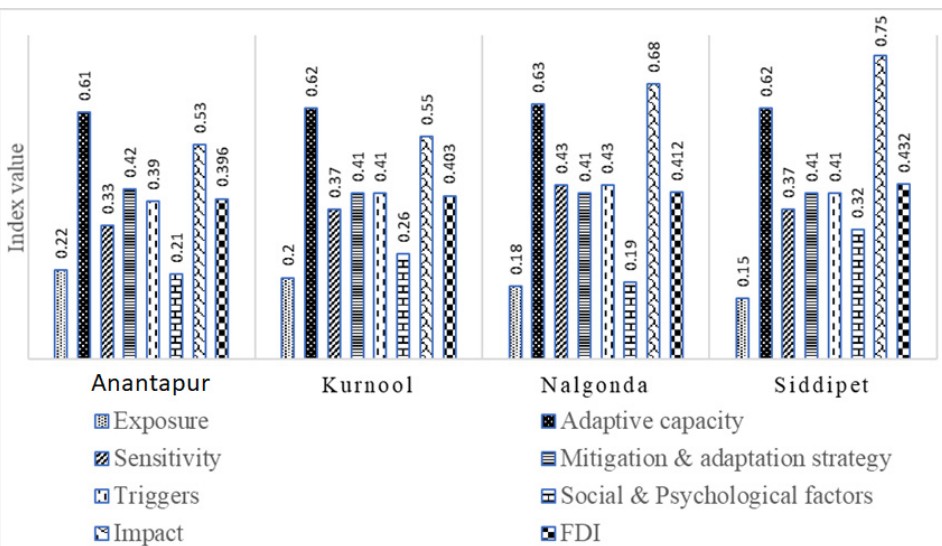

**Figure 13.** Dimensions of vulnerability and resulting Farmers' Distress Index (FDI) values for four districts.

As seen in Figure 13, the adaptive capacity and impacts contribute most significantly to the farmers' vulnerability. The demographic profiles that show a low educational status, high levels of caste discrimination, and a large disabled population increase the farmers' distress in the districts. Apart from these, having few economic assets, a large dependency on agriculture, a small landholding size, and a lack of irrigated area worsened the conditions. The sensitivity, triggers, and psychological factors had a moderate impact on farmers' distress. Finally, the dimension that had the lowest effect on vulnerability was exposure. Therefore, the overall result of the FDI suggests that the studied districts have moderate farmers' distress, and among all the districts, farmers in *Siddipet* face the highest distress. Nevertheless, a closer look into each dimension of vulnerability reveals some interesting differences. Overall, adaptive capacity has the lowest variation among the districts, possibly due to the similar agroecological situations (semi-arid tropics of India) and the similar historical and cultural backgrounds of the districts. At the same time, sensitivity, mitigation, and adaptation strategies, and triggers show moderate variation. The highest variability among the districts was reported for exposure, social and psychological factors, and impacts. Exposure was more pronounced in *Anantapur* district as it is historically known as a drought-prone area, while *Siddipet* showed a substantially lower index value; this indicates the greater vulnerability in *Anantapur* to droughts. Natural disasters such as droughts, prolonged dry spells, deepening of the water table, and borewell failures accounted for these differences (exposure table).

### 3.4. Utility of FDI

The above analysis of the principal dimensions and the contributing indicators of the dimensions of vulnerability can be used as a planning and monitoring tool to prioritize the districts with a high FDI to be considered for higher fund allocation [13,85,86] based on the extent of distress and also to help in the planning process by identifying which dimension needs to be prioritized for funding allocation and action by different stakeholders (Figure 14).

### 3.5. Scaling up to the Sub-District Level and Mapping of Vulnerability

The geographical areas of the districts in India are vast and heterogenous; climatic shocks such as droughts and floods are often localized and occur only in a specific part of the district rather than the entire district. Hence, there is a need for developing the FDI at the sub-district level (*mandal*/block: every district is sub-divided into *mandal*s/blocks in India, and each district comprises about 30–40 *mandal*s/blocks); within the *mandal*, the entire

area is homogenous. The sub-district is also the lowest level administrative unit for the majority of the government departments, such as agriculture, rural development, women and child development, and revenue, which makes it easy for different line departments of the government to make actionable decisions based on the FDI. Hence, the FDI was calculated at the *mandal* level as presented in Figure 15, depicting the different levels of severity of the farmers' distress with different symbols derived from the FDI. This mapping will be a powerful tool to identify clusters of high FDI scores and their dimensions [87,88].

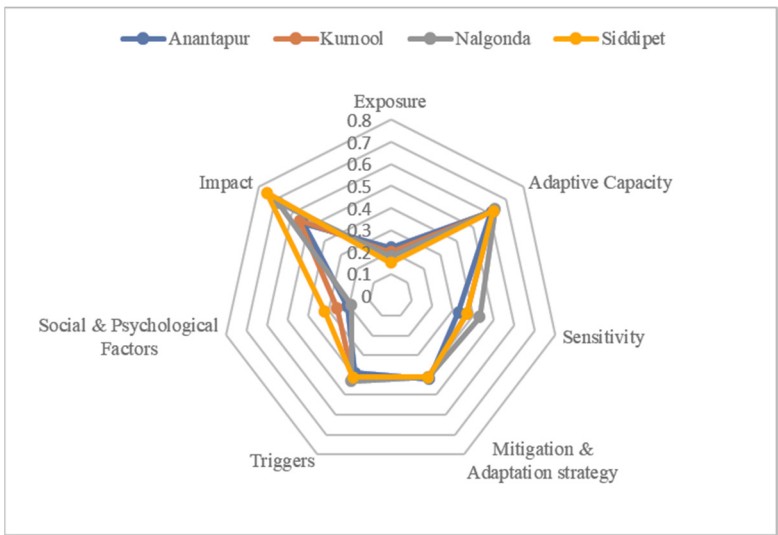

**Figure 14.** Vulnerability diagram for the major contributing components of the FDI.

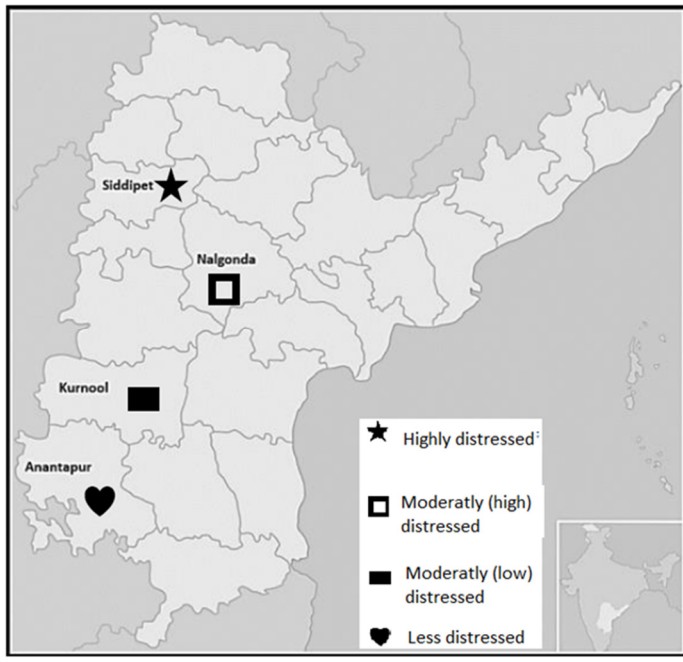

**Figure 15.** Mapping of severity of distress for the districts.

The *mandal*s/blocks of the districts were categorized into three groups (Table 5 and Figure 16) in terms of FDI values. It was revealed that within districts, there is a large degree of variability in the level of FDI. This verifies that, as a prioritization and planning tool, the FDI has to be measured at the *mandal*/block level to capture variability at the sub-district level [48,88–91].

**Table 5.** Prioritization of *mandal*s for future planning.

| District | *Mandal*s (Sub-Districts) in Different Levels of FDI | | |
|---|---|---|---|
| | Category A (High Distress) | Category B (Moderate Distress) | Category C (Low Distress) |
| Anantapur | | Kanganapalli | Ramagiri |
| Kurnool | Kudumuru | Pathikonda | |
| Nalgonda | Chandampet | Mariguda | Kattangur |
| Siddipet | Dubbaka, Markook | Mulugu | |

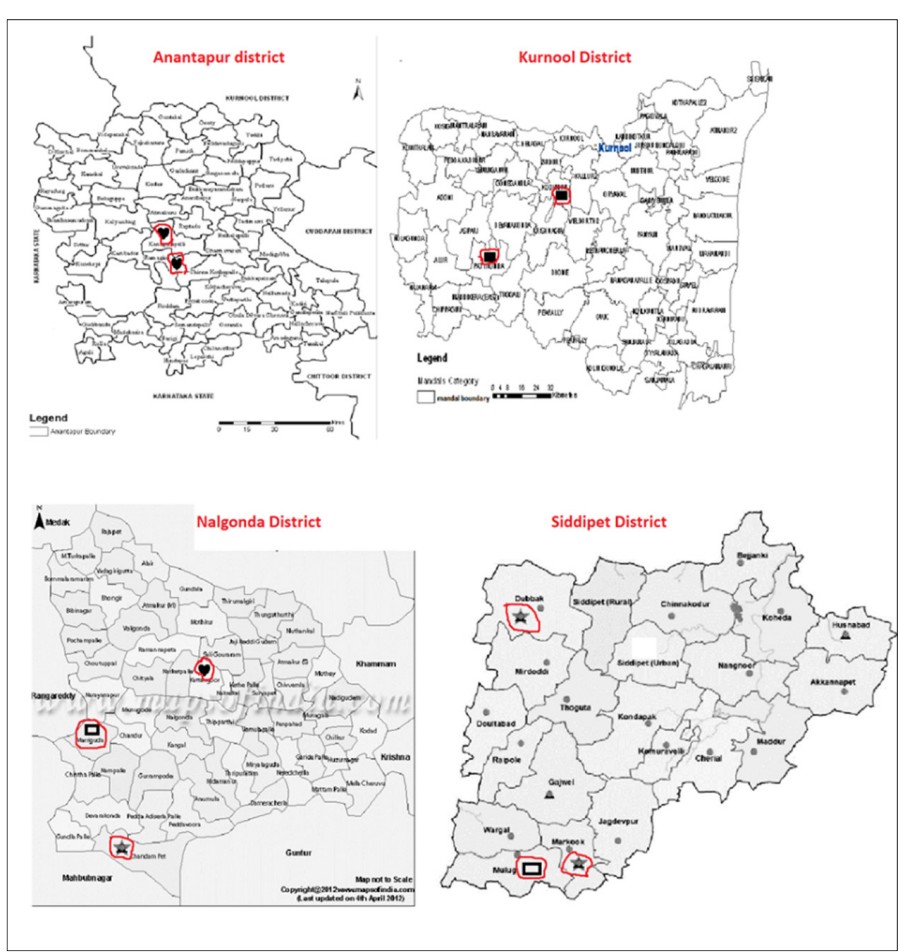

**Figure 16.** District maps with *mandal*s demarcated in red circles with respective categories based on the FDI.

a. Category A *mandal*s/blocks (demarcated as ★: Severe *mandal*/block distress (top 30% of the *mandal*s).
b. Category B *mandal*s/blocks (demarcated as □ for moderate–high and ■ for moderate–low): Moderate *mandal* distress (30 to 60% of the *mandal*s).
c. Category C *mandal*s (demarcated as ♥: Low *mandal* distress (bottom 40% of *mandal*s).

Dryland farmers, especially those in low-to-middle income countries, face many challenges, such as uncertain rainfall, prolonged dry spells, late onset of monsoons, rising production costs, outbreaks of natural disasters, biotic stresses such as diseases and pest attacks, fluctuations in market prices, etc. Combining all these variables into one composite FDI at the lowest level of the administrative unit and disseminating the information to all stakeholders will help with decision-making [92]. Accordingly, the current study measured the vulnerability of dryland farmers by developing the FDI at the *mandal* level. The results

revealed that agricultural vulnerability at the sub-district level is more beneficial for the prioritization and planning process as districts have a great deal of variability.

## 4. Conclusions and Future Work

### 4.1. Main Research Findings

The agricultural sector in India faces many challenges, such as rising demands, uncertainties regarding climatic changes, and natural disasters. Hence, updated information is essential to efficiently cope with climate-related risks and reduce agricultural distress [92,93]. Accordingly, the current study measured the agricultural distress of a particular study area by developing the FDI. The results revealed that agricultural distress varies across the locations. One important finding was that within each district, sub-district areas had different FDI scores; hence, from a policy point of view, using the FDI at the sub-district level as a prioritization and planning tool is essential to target the causes of farmers' distress [94] as this will help stakeholders to address the identified deficiencies and evolve measures to tackle them [95]. The FDI was relatively uneven across the sites, and accordingly, distress mapping was performed. Overall, we found that climate crisis conditions negatively affect the farmers' economic resources and lead to a profound disruption of social life within this community.

### 4.2. Policy Implications

The study results have implications for several policy areas in terms of tackling agricultural distress and preparing farmers to cope with the risks from exposure to climate change. Since India faces severe uncertainties in climatic changes, the small and marginal farmers have become extremely vulnerable. Therefore, considering these issues, the following policy recommendations are proposed. Firstly, social protection measures building on traditional risk diffusion measures should be proposed to improve the adaptive capacity of farmers. Secondly, policies that promote better access to crop insurance, weather-suitable crop variety, increasing awareness on water harvesting and conservation, and better access to weather information can play an essential role in increasing farmers' resilience. Finally, although the Indian government has allocated more resources to agriculture and several programs were initiated to improve the agricultural sector, agrarian distress is silently spreading across all the states. It seems that all these programs and schemes are disjointed and function independently of each other. Therefore, agrarian challenges and various ongoing programs should be brought together under one umbrella. This policy should cover the major issues such as increasing income, generating employment opportunities, reducing agrarian risks, developing agri-infrastructure, and improving the quality of rural life.

### 4.3. Research Challenges and Future Work

Although the present study has produced some significant and interesting results, there are certain research limitations and challenges. Due to the COVID-19 pandemic, the availability of consistent government data was affected, and it was time-consuming to conduct the field surveys in the villages and compile and finalize the data sets [96–98]. Likewise, due to the limited fieldwork time, the research could not be extended to more areas. In the era of climate change and post-liberalization, agricultural vulnerability and farmers' distress concern the entire society, including farmers, communities, policymakers, and researchers. Although several studies have focused on climate change dimensions and their resultant impacts on farmers' distress, a comprehensive and composite set of indicators representing all dimensions with great importance in farmers' distress that can be used as a policy tool is not appropriately addressed in the literature.

The study results have implications for several policy areas concerning agricultural distress and for preparing farmers and local administrations to cope with hazards through prioritization and planning at the sub-district (*mandal*/block) level. Although the selected 50 indicators in the seven dimensions are sufficient to diagnose the extent and duration of farmers' distress, combining the FDI with the latest satellite images will help to further

enhance the accuracy and utility of the FDI to allow timely actions to be taken before the realization of extreme distress. They can be obtained through remote sensing technology with minimal reliance on human intervention. It can also provide additional real-time data on many indicators such as soil moisture, temperature changes, biotic stresses such as the extent of pests and disease outbreaks, and yield assessment with more accuracy.

This improved FDI may be used to develop comprehensive agricultural insurance schemes, which have the potential to replace single-dimensional crop-specific insurance products, as insurance is one of the main policy instruments for reducing multidimensional farmers' distress [94]. The FDI offers a framework to evaluate and understand vulnerability at the farmer level. The FDI captures all aspects of farmers' livelihoods and vulnerabilities, including exposure to risk, sensitivity, adaptive capacity, mitigation and adaptation strategies, triggers, socio-psychological aspects, and impacts. Given that all aspects are covered with 50 simple indicators in seven dimensions, data can be collected from a representative sample of farmers from each sub-district (*mandal*) in identified vulnerable districts every year and based on the index, highly distressed *mandal*s can be identified and targeted for future policy intervention. The FDI tool can also work as an instrument to develop local community-driven climate resilience strategies through comprehensive bottom-up planning platforms, such as "Climate Innovation Platforms", that can be established in vulnerable districts throughout the country [95]. This analysis can also be usefully applied to study vulnerability patterns across other tropical regions of the world from a comparative perspective.

The FDI is the first step in developing a package of location-specific distress management approaches with periodical monitoring at every level to reduce farmers' distress. The process involves extensive knowledge transfer by the researchers to the policymakers to create a successful action plan for intervention through various organizations to provide various supports, inputs, and incentives (Figure 17).

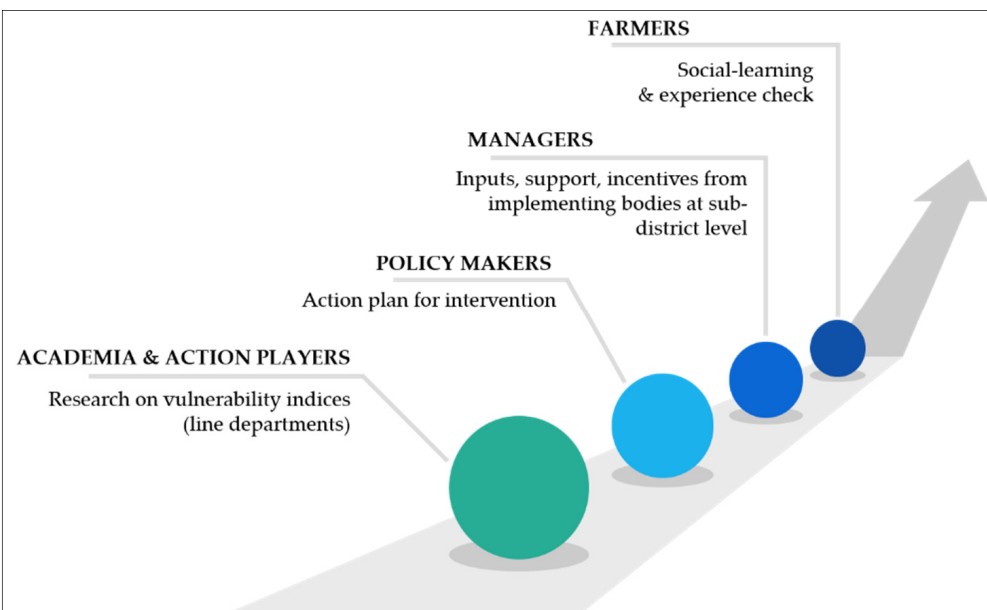

**Figure 17.** Diagrammatic representation of distress management package in agriculture (modified from [16]).

To properly implement the above package, developing a precise action plan with a separate budget allocation and an implementing agency is essential. We also emphasize the need to identify the bodies responsible for implementing the intervention plan, targeting each indicator at the sub-district level (Table 6).

**Table 6.** Action plan of distress management package (an example).

| Examples of Distress Indicators | *Mandal*s (Sub-Districts) Need Major Intervention | Area of Intervention | Action Plan for Intervention | Responsible Bodies |
|---|---|---|---|---|
| Inadequate total household income/higher share of agricultural income/lack of non-farm employment | Markook | Strengthening and training of small enterprises | Encouragement of women/youth in engaging in cottage industry with farm waste materials | NGOs and skill development council |
| Reduced income and high indebtedness | Kudumuru, Mariguda, Dubbaka, Murugu | Credit support | Easy and smooth access of formal credit institution | Formal credit institutions (banks, cooperatives) |
| Low household assets value | Dubbaka, Markook, Murugu | | Monitoring and proper delivery of various asset-generating schemes such as housing, vehicles, or agricultural implementations | Local-level government bodies (Panchayat) |
| High farm expenses | Dubbaka, Mariguda | Cost-effective technology, subsidies, improving yields | Proactiveness of government with agricultural subsidy schemes to reach vulnerable farmers Farm mechanization Usage of renewable resources | State-level government body Research institutions, NGOs Dept. of agriculture |
| Income from agriculture | Pathikonda, Dubbaka, Kudumuru | Guaranteed support price, value addition | Effective implementation of minimum support price (MSP) Banks/insurance Small-scale processing plants | Local-level government bodies (Panchayat) Department of agriculture Income insurance |
| Low agricultural landholding | Kanganapalli, Pathikonda | Mapping of local resources and their management, development of land-lease markets | Adoption of integrated farming system, credit facilities to tenant farming | Local bodies Banks |
| Failure of borewell (lack of irrigation) | Chandampet, Kudumuru | Watershed development | Practice of water harvesting and conservation | Local bodies Community Extension agencies |
| Low educational status | Dubbaka | Capacity building | Provision of extension services and special training to farmers | Research institutions NGOs, SHG |
| Crop failure | Kanganapalli, Mariguda, Markook | Promotion of involvement in the mitigation program | Awareness of crop insurance schemes Identification of drought-prone areas Selection of proper crop varieties | Local-, state-, and national-level government bodies Insurance companies |

Additionally, this paper also provides a conceptual model of the Distress Man-agement Package at the sub-district level to develop a network between various play-ers and develop a location-specific action plan to mitigate agricultural distress [27,99,100]. The FDI can be used as a policy tool, especially in states with highly recur-rent farmer distress such as Maharashtra, Telangana, Karnataka, and Rajasthan, with regular field surveys conducted using the identified 50 variables for farmers to con-struct a sub-district-level FDI and to categorize and prioritize action points by the gov-ernment and the local community to reduce farmers' distress. This can trigger virtuous social innovation and represents a new frontier of sustainable and resilient develop-ment through an effective communication system to reduce agrarian distress.

**Supplementary Materials:** The following are available online at https://www.mdpi.com/article/10.3390/land10111236/s1, Table S1: Rating of variables of indicators, Table S2: Indices of crop loss during last 5 years due to natural hazards (Exposure to risk), Table S3: Indices of socio-demographic profile, Table S4: Indices of socioeconomic assets, Table S5: Indices of livelihood strategies, Table S6: Indices of social networks, Table S7: Indices of agricultural activities, Table S8: Indices of sensitivity indicators, Table S9: Indices of farmers' own initiative towards mitigation strategies, Table S10: In-dices of government support, Table S11: Indices of adaptation strategies to reduce distress, Table S12: Indices of the constraints in adaptation measures, Table S13: Indices of triggers, Table S14: Indices of

social and psychological factors causing agrarian distress, Table S15: Indices of the impact of agrarian distress on farmers' livelihoods.

**Author Contributions:** Conceptualization, A.A.R.; methodology, A.A.R.; software, A.A.R.; validation, A.A.R. and A.B.; formal analysis, A.A.R., A.B. and S.R.; investigation, S.V.R.; resources, S.V.R.; data curation, A.A.R.; writing—original draft preparation, A.B.; writing—review and editing, A.A.R., A.B. and S.R.; visualization, A.A.R., A.B. and S.R.; supervision, S.R. All authors have read and agreed to the published version of the manuscript.

**Funding:** This research received no external funding.

**Data Availability Statement:** Dataset is available on request to authors.

**Acknowledgments:** We acknowledge the technical support received from DVARA foundation.

**Conflicts of Interest:** The authors declare no conflict of interest.

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
