# Peer review of "Farmers’ Distress Index: An Approach for an Action Plan to Reduce Vulnerability in the Drylands of India"

_land, doi:10.3390/land10111236_

Round 1

Reviewer 1 Report

The handled paper aims to produce a new compound index on farmers' distress vulnerability in Indian drylands. The paper is a good job, it reads well and is of interest for Land. The literature considered is abundant and pertinent, the topic worthy of publication. The abstract can be shortened. Prefer writing the numbers in words, for example two instead of 2. Some papers in the reference list are not cited, see for example Tate, 2013. Along with Tate, 2013, the authors may use additional papers to motivate this topic. Check the manuscript typos. There is a typo within the title to be corrected. Authors contributions are currently underlined in yellow. I suggest approving the paper after this.

Author Response

Farmers’ Distress Index: An approach for an action plan to reduce vulnerability in the drylands of India

The authors would like to thank all the reviewers for their comments and suggestions. We consider that the suggestions, and our subsequent amendments, have strengthened the manuscript. Attached is a description of the amendments made by the authors.

Addressing the comments of Reviewer 1:

Point 1. The abstract can be shortened.

The abstract has been shortened.

Point 2. Prefer writing the numbers in words, for example two instead of 2.

Some of the numbers have been written in words. But some remained in numerical for easy understanding.

Point 3. Some papers in the reference list are not cited, see for example Tate, 2013.

The reference list has been checked and the non-cited papers have been removed from the list.

  1. B. Tate, 2013 (listed in 79 in reference) is mentioned in the text in line no. 571-575 (“The above analysis of the major dimensions and the contributing indicators of the dimensions of vulnerability can be used as a planning and monitoring tool to prioritize the districts with a high FDI to be considered for higher fund allocation based on the depth of distress and also help in the planning process by identifying which dimension needs priority in funding allocation and action by different stakeholders (Figure 13; 12, 77-80”].)

Point 4.  Along with Tate, 2013, the authors may use additional papers to motivate this topic.

Apart from Tate, 2013, Hofman, 2018, The World Bank, 2006 and Salazar-Briones, 2020 are already mentioned to support this topic.

Point 5. Check the manuscript typos. There is a typo within the title to be corrected.

It has been corrected.

Point 6. Authors contributions are currently underlined in yellow.

It has been corrected.

Reviewer 2 Report

The subject of the article is interesting, and it is linked to the objectives of the journal, however, there are many issues that have to be reconsidered.

Abstract is too long, I suggest to make it shorter and more precised. 

For better visibility on databases, the authors are asked not to repeat among keywords the words/concepts included in the title of the article.

The part called "Objectives of the study", seems to be more a mathodological part, and it does not ireflect the title of the sub-chapter. It must be reconsidered.

A map of the region could help understanding the research area.

I reccommed a dedidacted section of Conclusions, where to answer to the questions rised during the research. 

Author Response

land- 1454389

Farmers’ Distress Index: An approach for an action plan to reduce vulnerability in the drylands of India

The authors would like to thank all the reviewers for their comments and suggestions. We consider that the suggestions, and our subsequent amendments, have strengthened the manuscript. Attached is a description of the amendments made by the authors.

Addressing the comments of Reviewer 2:

Point 1. Abstract is too long, I suggest to make it shorter and more precised. 

The abstract has been shortened.

Point 2. For better visibility on databases, the authors are asked not to repeat among keywords the words/concepts included in the title of the article.

Repeated words (or concept) have been removed from the keywords.

Point 3. The part called "Objectives of the study", seems to be more a methodological part, and it does not reflect the title of the sub-chapter. It must be reconsidered.

The above sub-chapter has been reconstructed as per the reviewer’s comment.

Point 4. A map of the region could help understanding the research area.

Map has been added.

Point 5. I recommend a dedicated section of Conclusions, where to answer to the questions raised during the research. 

The Conclusions section highlights the main results and provides some reflection on future works. Although we are aware that this section does not follow a usual style, we think that it could provide useful “go forward” issues and motivations to deepen conceptual models to motivate further research.

Reviewer 3 Report

I appreciate the authors’ revisions to improve the manuscript. Most of my remarks from the previous round of revision has been addressed. However, I believe some issues need revision and clarification.

Author Response

land- 1454389

Farmers’ Distress Index: An approach for an action plan to reduce vulnerability in the drylands of India

The authors would like to thank all the reviewers for their comments and suggestions. We consider that the suggestions, and our subsequent amendments, have strengthened the manuscript. Attached is a description of the amendments made by the authors.

Addressing the comments of Reviewer 3:

Point 1: Lines 23-25 in page 1: “Focusing on this objective, the present paper attempts to build an FDI with seven dimensions of distress: exposure to risk, adaptive capacity, sensitivity, mitigation and adaptation strategies, triggers, psychological factors, and impacts.ˮ Does this sentence refer to the main aim of this study? The authors should mention the main aim of their study in the Abstract clearly.

The abstract has been reformulated and the aim of the study has been indicated clearly.

Point 2:  Lines 26-28 in page 1: “At first 123 indicators through participatory rural appraisal techniques, then screened all the indicators for suitability and finally selected only 50 indicators based on the pilot test.ˮ The authors should reword and reformulate this sentence.

The entire abstract including this sentence has been reformulated.

Point 3: Lines 33-36 in page 1: “Hence, the study concludes by explaining the FDI as a planning tool that can address the causes of farmers’ distress and evolve measures to tackle them.ˮ The authors should reword and edit this sentence for more fluency.

The sentence has been edited.

Point 4: Lines 37-39 in page 1: “It should involve the local administration, research institutions and NGOs to prioritize the allocation of funds and build specific action plans according to the dimensions of the FDI in each sub-district.ˮ The underlined part is vague and needs further explanations.

As the entire abstract has been reformulated, this part has been edited and discussed in the main text.

Point 5: Lines 39-43 in page 1: “Lastly, the paper suggests an upscaling policy of building a five point program (increasing income, generating employment opportunities, reducing agrarian risks, developing agri-infrastructure and improving quality of rural life) that addresses agrarian challenges and bring together various ongoing programs under one 2 umbrella.ˮ is too long and heavy; the authors should reword, reformulate and split this sentence.

The sentence has been reformulated.

Point 6. Lines 59-63 in page 2: “After the process of liberalization i.e. relaxation of government regulations and restrictions for greater participation by private entities since the 1990s, rainfed farmers are becoming further exposed to price risks as they heavily depend on markets for purchasing inputs and also selling outputs, unlike subsistence farming [8].ˮ The authors should reformulate split this sentence.

The sentence has been reformulated.

Point 7. Lines 59-63 in page 2: “The challenges faced by rainfed agriculture are interrelated with the Sustainable Development Goals of no poverty (SDG- no.1), zero hunger (SDG- no.2), climate action (SDG- no. 13) and life on land (SDG- no.15).ˮ In this sentence, the authors should mention that these abbreviations with specific codes are related to the sustainable development goals established by the United Nations in 2015 and add the relevant reference.

The sentence has been reformulated and the relevant reference has been added.

 Point 8. Lines 79-82 in page 2: “Earlier studies [12] (e.g., ....) observed that the severe crisis in dryland agriculture in the past century has resulted in increased levels of poverty and low investment in both human and physical capital, which reinforce low agricultural productivity and low incomes.ˮ The authors should mention more than one study and correct the citation (it should be Krishnamurthy [12]). Please address my comment in the underlined part.

The corrections have been done as per the suggestion.

Point 9. Line 181 in page 5: “The study not only develops an FDI but also decomposes the index into seven dimensions.ˮ What do the authors mean by decomposesˮ in the sentence? I would suggest replacing it with analysis.

Corrected. Thank you.

Point 10. Comments in the Methodology section have been addressed carefully.

Thank you.

Point 11. Lines 359-360 in page 11: “Several studies (e.g., …) elucidated that the preponderance of marginal holdings is increasing due to great demographic pressure and land segregation.ˮ The authors should mention which studies obtain these outcomes (please complete the underlined part).

The sentence has been completed as per the suggestion.

Point 12. Lines 360-361 in page 11: “In the present study, the average operational land-holding of the sample households was 3.8 acres (1.52 hectare), among which 90% of the area was under rainfed farming.ˮ What is the areal unit of acres? I would suggest indicating the conversion between acres and ha in parenthesis.

Conversion has been mentioned.

Point 13. The authors should use three-line format for tables.

Table format has been changed.

Point 14. Lines 423-429 in page 14: “This indicator represents the households’ capacity to cope up with distress conditions. Socio-demographic profiles (which includes variables such as caste, sex ratio of family, educational status of head, dependency ratio), socio-economic assets (house value, total own land), livelihood strategies (total savings, Simpson Income Index, Simpson Diversity Index—cropping pattern), social networks (membership in SHGs/Cooperatives), agricultural activities (income from agriculture, profit and rainfed area) are components of the adaptive capacity index.ˮ This sentence is too long and heavy; the authors should reformulate and split it.

The sentence has been reformulated.

Point 15. Lines 430-435 in page 14: “Caste (forward caste = 2; backward castes = 1; scheduled castes = 0) and the educational status of the head of the household (years of education) were assumed to be inversely associated with FDI among socio-demographic variables, while among livelihoods strategies, total savings, the Simpson Income Index and Simpson Diversity Index (SDI) of cropping patterns were assumed to have an inverse association with FDI based on the focus group discussions.ˮ again, too long sentence, please split and reformulate this sentence.

The sentence has been splitted.

Point 16. Lines 493-497 in page 16: “Most farmers are beneficiaries of government schemes, and almost all landholders benefited through the Rythu Bandhu scheme (which is a direct benefit transfer program by the government to transfer .10,000/acre/annum); some farmers are beneficiaries of an old age pension scheme through an insurance scheme, and some participate in employment guarantee schemes such as MGNREGA, etc.ˮ This 4 sentence is too long due to the wrong use of semicolon. Moreover, why some words are italic and capital? The authors should modify them.

The sentence has been edited as per the suggestion. The “Rythu Bandhu” word are italic since it is a local term, not an English word. “MGNREGA” is in capital because it is an abbreviation. The full form of this has been mentioned in parenthesis.

Point 17. The authors should separate Discussion and Conclusion section.

Conclusion part has been separated.

Point 18. Figure 6 and Table 15 with their explanations should be moved to the Results section.

Figure 6 is already in result section and Table 15 is not mentioned in the text.

Point 19. The Discussion section just repeated the results of the study. The findings and their implications should be discussed in the broadest context possible and limitations of the work.

The findings are already discussed in “main research finding” sub section and the limitations of the work are also mentioned in “research challenges” sub-section.

Point 20. In Conclusion section, the authors should discuss the main implication of the findings and the importance of the results.

The main implications of the findings are included in the updated “Conclusion and future work” section by indicating an action plan of Distress Management Package.

Round 2

Reviewer 2 Report

Dear author/s,

Thank you for the improved version of the manuscript.

Hood luck with your future research!

This manuscript is a resubmission of an earlier submission. The following is a list of the peer review reports and author responses from that submission.

Round 1

Reviewer 1 Report

Thanks for this invitation. The submission contributes enriching the literature of sustainability synthetic indicators. Most of the sections are solid. In my opinion, there are just a few corrections to be made. I suggest securing the following:

  • The connection with land research is evident. Though you can specify the connection with the paper in the introduction.

  • Composite indicators need uncertainty analysis. You will have to perform a robustness test, check: 

1) A Robust Approach to Composite Indicators Exploiting Interval Data: The Interval-Valued Global Gender Gap Index (IGGGI), 2018.

2) Interval Based Composite Indicators, 2017.

3) On the Methodological Framework of Composite Indices: A Review of the Issues of Weighting, Aggregation, and Robustness, 2019.

  • Ensure a native language speaker/professional proofreading. The paper is of good quality, it would be a shame to present syntax and grammar mistakes.

  • What is the "adoptive" capacity? I think you mean "adaptive" capacitity, as a resilient response to change.

  • Has this paper any flaws? What following steps are needed in this stream of research. You should explain in the conclusion.

Reviewer 2 Report

Dear author/s,

the topic of the paper is interesting can the results could be helpful for future researchers.

In the introduction section are presented the main research directions regarding farmers' index of vulnerability. However for a better understanding of the topic I recommend to systemize the main researches in a table, being easier to underline the novelty of the research. I also consider it would be good be presented in the introduction/literature review section the methods and results of other similar studies (e.g. Suryanto and Rahman, 2019; Kantamani et al., 2020 etc.) some of them being mentioned in the results part.

The discussions part is too poor at this moment. Please try to improve it and compare more your results with other similar results.

Good luck!

Reviewer 3 Report

This paper investigated Farmers’ Distress Index (FDI) with seven dimensions of distress to examine the development of composite index to track farmers’ distress at the sub-district level to take action for sustainable living. Overall, this study addresses a topic of high relevance for research and also for practice. However, I believe in this paper, some issues need revision and clarification.
